

# Image based quantitative comparisons indicate heightened megabenthos diversity and abundance at a site of weak hydrocarbon seepage in the southwestern Barents Sea

Arunima Sen[1,5], Cheshtaa Chitkara[1], Wei-Li Hong[2], Aivo Lepland[2], Sabine Cochrane[3], Rolando di Primio[4] and Harald Brunstad[4]

[1] UiT The Arctic University of Norway, Centre for Arctic Gas Hydrate, Environment and Climate (CAGE), Tromsø, Norway
[2] Geological Survey of Norway (NGU), Trondheim, Norway
[3] Akvaplan-niva, High North Research Centre, Tromsø, Norway
[4] Lundin Norway, Oslo, Norway
[5] Current affiliation: Faculty of Biosciences and Aquaculture, Nord University, Bodø, Norway

Corresponding author
Arunima Sen, arunima.sen@nord.no,
borshusen@gmail.com

## ABSTRACT

**Background**. High primary productivity in the midst of high toxicity defines hydrocarbon seeps; this feature usually results in significantly higher biomass, but in lower diversity communities at seeps rather than in the surrounding non-seep benthos. Qualitative estimates indicate that this dichotomy does not necessarily hold true in high latitude regions with respect to megafauna. Instead, high latitude seeps appear to function as local hotspots of both megafaunal diversity and abundance, although quantitative studies do not exist. In this study, we tested this hypothesis quantitatively by comparing georeferenced seafloor mosaics of a seep in the southwestern Barents Sea with the adjacent non-seep seafloor.

**Methods**. Seafloor images of the Svanefjell seep site and the adjacent non seep-influenced background seabed in the southwestern Barents Sea were used to construct georeferenced mosaics. All megafauna were enumerated and mapped on these mosaics and comparisons of the communities at the seep site and the non-seep background site were compared. Sediment push cores were taken in order to assess the sediment geochemical environment.

**Results**. Taxonomic richness and abundance were both considerably higher at the seep site than the non-seep location. However, taxa were fewer at the seep site compared to other seeps in the Barents Sea or the Arctic, which is likely due to the Svanefjell seep site exhibiting relatively low seepage rates (and correspondingly less chemosynthesis based primary production). Crusts of seep carbonates account for the higher diversity of the seep site compared to the background site, since most animals were either colonizing crust surfaces or using them for shelter or coverage. Our results indicate that seeps in northern latitudes can enhance local benthic diversity and this effect can take place even with weak seepage. Since crusts of seep carbonates account for most of the aggregating effect of sites experiencing moderate/weak seepage such as the study site, this means that the ability of seep sites to attract benthic species extends well beyond the life cycle

of the seep itself, which has important implications for the larger marine ecosystem and its management policies.

## INTRODUCTION

The enrichment of subsurface fluids in reduced compounds and hydrocarbons (most notably sulfide and methane), and their seepage at the seafloor creates ecosystems known as cold seeps. Tectonic activity, transformation of organic deposits, gas hydrate dissociation and sub-seabed movement of salt formations are among some of the processes that lead to the expulsion of cold seep fluids (*Sibuet & Olu-Le Roy, 1998*; *Sibuet & Olu-Le Roy, 2002*; *Cordes, Bergquist & Fisher, 2009*; *Levin et al., 2016*). The reduced seeping compounds provide substrates for both aerobic and anaerobic microbial oxidation activities. These processes can be linked to carbon fixation (known as chemosynthesis) and primary production can take place at cold seeps regardless of whether sunlight is available and photosynthesis is possible. Certain kinds of chemoautotrophic bacteria bond symbiotically with animals, including at the intracellular level, leading to large bodied animals essentially functioning as primary producers within the cold seep ecosystem (*Dubilier, Bergin & Lott, 2008*). Subsequently, cold seeps are often biomass-rich locations on the seafloor, and particularly conspicuous in the deep sea or below the photic zone, where fauna tend to be sparse due to their reliance on low quantities of photosynthesis based material, slowly descending through the water column (*Fisher et al., 2007*; *Kiel, 2010*; *Levin et al., 2016*).

The caveat to life in cold seep oases is that reduced fluids, such as sulfide, are highly toxic: they inhibit oxygen transport and block oxidative respiration (*Beauchamp Jr et al., 1984*; *Bagarinao, 1992*; *Truong et al., 2006*). Therefore, cold seeps usually host a small number of species or taxa, which, however, can be very abundant (*Sibuet & Olu-Le Roy, 1998*; *Sibuet & Olu-Le Roy, 2002*; *Levin et al., 2016*). In other words, if taxa can solve the issue of sulfide poisoning, they can flourish at cold seeps. As a result, cold seeps have been seen to host high biomass, but low diversity communities, compared to the surrounding non-seep seafloor (*Sibuet & Olu-Le Roy, 1998*; *Sibuet & Olu-Le Roy, 2002*; *Fisher et al., 2007*; *Kiel, 2010*; *Levin et al., 2016*). However, as studies on cold seeps expand and entire new regions are being described, the high-biomass/abundance, but low diversity paradigm of seeps is coming into question (*Sellanes et al., 2010*). In particular, seeps in high latitude regions appear to host both high abundance and high diversity in comparison to surrounding non-seep areas. *Åström et al. (2018)* demonstrated this quantitatively with respect to sediment infaunal communities. At the megafaunal scale (i.e., animals at least one cm in dimension and visible to the naked eye (*MacDonald et al., 2010*; *Amon et al., 2017*)), a similar pattern has been suggested, but quantitative studies have not been conducted to confirm this trend (*Gebruk et al., 2003*; *Sen et al., 2018b*).

Our goal was to address this gap, and towards this end, we carried out a quantitative comparison between the megafaunal communities of a cold seep and the nearby non-seep

background area, in the southwestern Barents Sea. The aim was to determine whether both faunal abundance and diversity (as opposed to simply the former) are higher at a high latitude cold seep location in comparison to the surrounding seafloor. The site location is in the area of potential hydrocarbon exploitation (the Svanefjell prospect), therefore our results have both theoretical and practical significance. As our knowledge of chemosynthesis based systems such as cold seeps has expanded, so has our understanding of the significance of such habitats to the larger marine ecosystem. Even influences that appear to be highly localized, such as the provision of food or shelter, in fact, have large scale and even planetwide footprints, such as impacts on global geochemical cycles and nutrient cycling (*Le Bris et al., 2017*; *Levin et al., 2016*). An initial step of quantifying and comparing previously unstudied seeps to the background seafloor can therefore offer considerable insight on benthic community dynamics. High resolution imagery and mapping within a Geographic Information System (GIS) was combined with sediment porewater geochemical measurements to gain an understanding of how physical, abiotic features relate to the biological communities at the Svanefjell site.

## MATERIALS AND METHODS

The Svanefjell seep site is located on the continental shelf of the Barents Sea off the coast of Hammerfest, northern Norway, at a water depth of about 380 m (72°4′N, 21°48′E, Fig. 1). Anomalies, indicating the presence of gas in subsurface sedimentary strata in the area, were identified in 3D seismic profiles. The site itself was identified based on high-resolution multibeam echosounder surveys conducted by Lundin Norway where a gas flare in the water column was observed (*Brunstad, Andersson & Pedersen, 2016*). The multibeam survey results also revealed the occurrence of numerous 30 to 50 m diameter pockmarks—circular depressions formed due to fluid seepage on the seafloor (Fig. 1). The gas flare observed via the echosounder was located within one of these pockmarks and visual observations of the seafloor within the central part of the pockmark using an ROV (remotely operated vehicle) identified bubbling gas escaping from the seafloor in that location. In April 2018, the Svanefjell site was visited with the anchor-handling vessel M/S *Bourbon Arctic* with the approval of Lundin Norway (Production License 934). The pockmark where the gas flare was observed was singled out for detailed study. It consists of a relatively large, 80 m wide and 2 m deep pockmark with a central high (1m relative height) (Fig. 1). Though visual observations during the mosaicking surveys (below) did not reveal any bubbles in 2018, abundant crusts of authigenic seep carbonates, i.e., another feature that is typical for a cold seep setting (*Crémière et al., 2016b*; *Chand et al., 2017*), were seen within the pockmark.

The working-class ROV on board the vessel was used to image the seep in the central part of the pockmark. A vertical, downward facing video camera (TechnipFMC Schilling Robotics high definition camera) was mounted to the bottom of the ROV and the ROV was flown slowly at a speed of about 0.5 knots (900 m/s) in a lawn mower fashion, to provide complete coverage, ensuring overlap between successive lines. Altitude was maintained at about 2 m throughout the imaging process. Images were extracted from the video every five seconds with the free software FFmpeg (http://ffmpeg.org/) and time stamps were used to

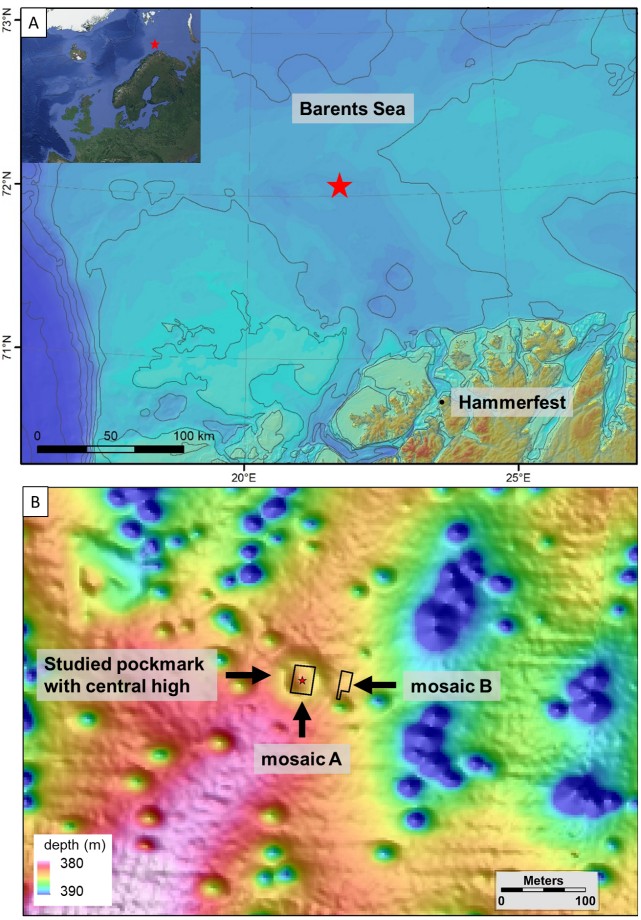

**Figure 1 Map of the study site.** Map and location of the Svanefjell site. The inset and (A) shows the location of the Svanefjell site in relation to northern Norway. The red stars indicate the site. (B) is a bathymetric map of the Svanefjell site. The crater or pockmark like feature with a central high where mosaic A was made and push cores were taken is highlighted. The outlines of both the mosaics are shown. Within the study pockmark, the star marks the location where a gas flare was observed on the echosounder during a prior cruise. Satellite imagery for the inset was taken from Google Maps (Data: SIO, NOAA, US Navy, NGA, GEBCO IBCAO, Landsat/Copernicus US Geological Survey).

link corresponding navigation data from the ROV. Images and navigation files were then inputted into the PhotoScan software (Agisoft, version 1.3.4 build 5067, 2017) to construct a georeferenced mosaic (mosaic A). About 30 m away from the seep, the same procedure was repeated in order to construct a mosaic of the background seafloor (mosaic B).

Both mosaics were imported into ArcMap 10.5 and all visible features were manually marked via the Editor tool. Every individual of the different faunal groups was marked and quantified (point feature class). General classifications, based on visible morphology were used, since most of the taxa seen in the mosaics are difficult to identify to species level through imagery. Therefore our faunal lists consist largely of morphotaxa and genera. Since the two mosaics covered different areas of the seafloor, densities of each taxon were calculated based on the spatial extents of the mosaics, to allow for comparisons to be made.

In addition to quantifying individuals of the different taxa, aggregations of unidentifiable organisms in the sediment were seen and these were outlined as polygons (polygon feature class in Arc). Microbial mats, crusts of seep carbonates and stones (cobble and boulder size rock clasts) were marked and quantified as polygon feature classes as well. Cod (*Gadus morhua*) were abundant at both mosaicking locations, however, their mobility makes them difficult to enumerate since it is impossible to differentiate between newly visible individuals and those that had previously been marked. Therefore individuals of this species were not marked in the mosaics, but it should be kept in mind that they were present in large numbers within both mosaicked areas.

Four push cores were taken from the seep next to crusts of seep carbonates, in order to obtain an overview of the geochemical characteristics of the seep site (Fig. 2). Sulfide ($\Sigma HS = HS^- + S^2$) and sulfate ($SO_4^{2-}$) concentrations were measured along the length of the cores. Porewater was extracted in a temperature-controlled room (4 °C) with acid-washed rhizon samplers and syringes, and 0.5 to 2 ml of porewater was preserved with $Zn(OAc)_2$ onboard (<30 min after rhizons were disconnected) for further analyses in the lab. Samples were kept frozen all the time upon analyses. Measurements were made every few cm, with the first measurement usually having been made at the sediment-water interface. We used the 'Cline method' to determine sulfide concentration (*Cline, 1969*). For most of the samples, two replicated measurements were performed. Uncertainty of the analysis for each sample was calculated from these replicates. The detection limit of sulfide was 0.2 µM. Sulfate concentration was determined using a Dionex ICS-1100 Ion Chromatograph (IC) with a Dionex As-DV autosampler and a Dionex IonPac As23 column (eluent: 4.5 mM $Na_2CO_3$/0.8 mM $NaHCO_3$, flow: 1 ml/min). The relative standard deviations from repeated measurements of different laboratory standards are better than 0.5% for concentrations above 0.1 mM. The detection limit for sulfate is ca. 0.01 mM.

## RESULTS

Cnidarians, sponges, seastars and fish were the only living taxa seen in mosaic A (Figs. 2–3, Table 1), while mosaic B (non-seep background) contained only sponges and one individual of *Sebastes* rockfish (plus cod,). Merely 19 individuals of the sponges were seen in mosaic B, therefore they were present in much lower numbers in the non-seep area than in mosaic A within the seep area (269 individuals). In both mosaics, some areas contained small, white objects in the sediment that could not be identified, but are possibly small sponges, cnidarians or polychaetes (or parts of them, Fig. 4). The patches of small, unidentifiable white animals were more widespread in mosaic A than mosaic B. Holes in the sediment were observed in similar quantities in both mosaics (densities of 2.06 and 2.02 per $m^{-2}$ in mosaic A and B respectively, Table 1). The holes alone cannot be used to make any definitive identifications, however, based on their size and sightings from nearby locations, it is possible that they are inhabited by amphipods, polychaetes, shrimp or bivalves. Therefore, at least one other megafaunal species, and one other megafaunal phylum is present in the background area, and that too, at similar densities as the seep location. Nonetheless, mosaic B clearly has much fewer taxa and overall, much fewer individuals (abundance) than mosaic A (Figs. 2–3, Table 1).
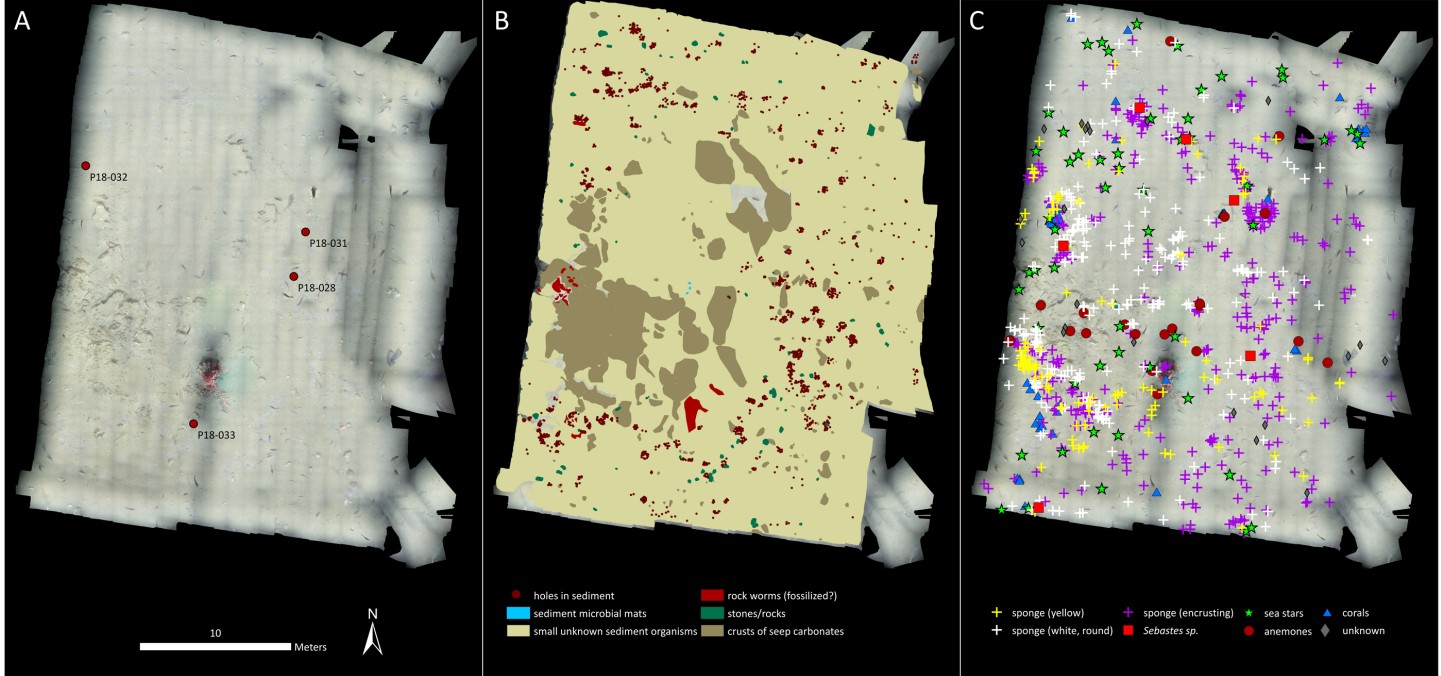

**Figure 2** **Mosaic A.** (A) The georeferenced mosaic with the locations where push cores were taken marked. (B) The mosaic with polygon categories shown, plus the locations of the holes in the sediment. (C) The mosaic with all living fauna (point categories) shown. Note that cod were not marked.

Surprisingly, siboglinid polychaete worms were absent from the seep site, despite this group constituting the dominant megafaunal group at high latitude seeps in the Atlantic and Arctic Oceans (*Gebruk et al., 2003*; *Rybakova Goroslavskaya et al., 2013*; *Paull et al., 2015*; *Åström et al., 2016a*; *Åström et al., 2018*; *Sen et al., 2018a*; *Sen et al., 2018b*). Similarly, sediment microbial mats were mostly absent at the seep; only three, small mats were visible in mosaic A. However, crusts of seep carbonates were present and accounted for about 13% of the area within mosaic A. Within or on some of the crusts, possible fossilized worms were visible, though an exact identification was not possible (Fig. 5).

Stones and rocks constituted another source of hard substrate in the mosaicked areas, but appeared to be much smaller in size overall in comparison to crusts of seep carbonates. We attempted to differentiate between crusts and stones in the mosaics since they have different textures that are often visible in images. Despite this, our characterizations might not be 100% accurate, but nonetheless, for the most part, we were able to mark and quantify both crusts and stones/rocks in the two mosaics. Based on this, we determined that the average size of stones in mosaic A was 0.04 m$^2$ and 0.008 m$^2$ in mosaic B, while the average size of crusts was 0.6 m$^2$. Therefore, crusts were considerably larger than stones and this is purely from a 2D perspective and does not even include their height above the seafloor. In mosaic A, none of the anemones, and only five individuals of corals (out of 52), five individuals of encrusting sponges (out of 361) and one individual each of yellow sponges (out of a total of 111) and white, rounded sponges (out of 269 individuals) were located on

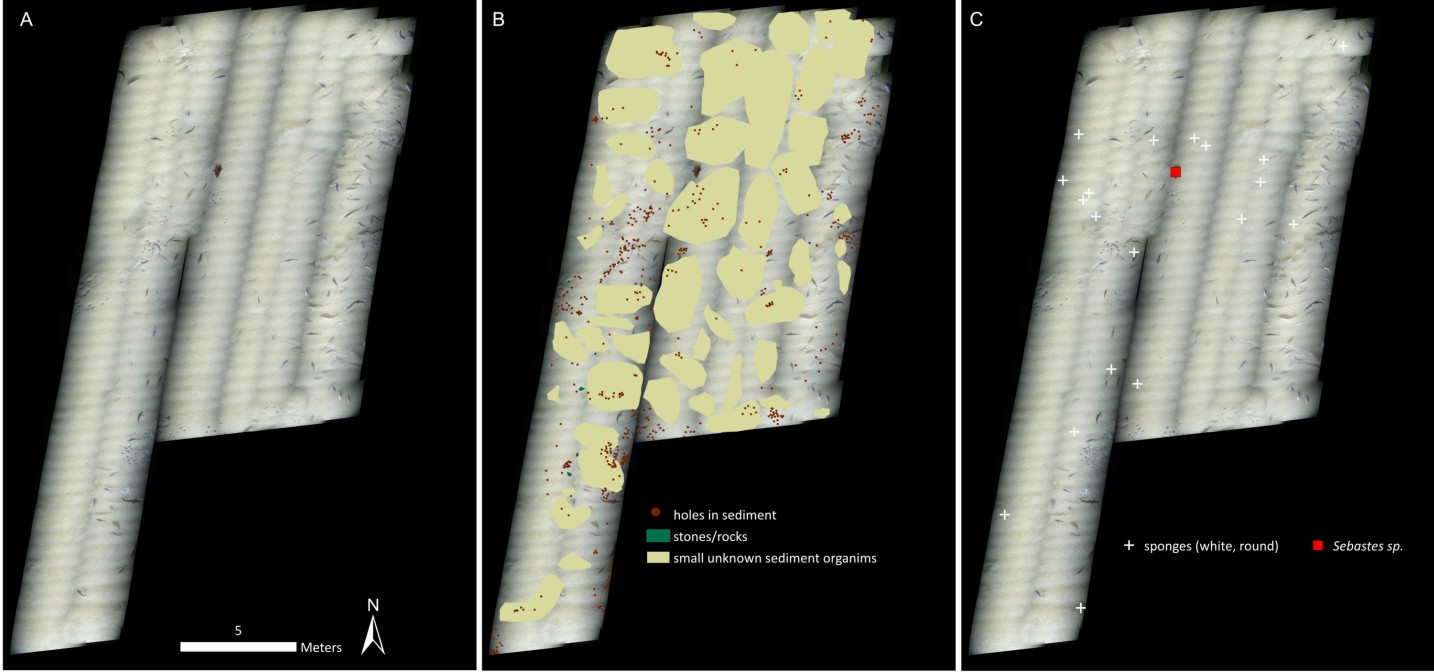

**Figure 3 Mosaic B.** (A) The georeferenced mosaic. (B) The mosaic with polygon categories shown, plus the locations of the holes in the sediment. (C) The mosaic with all living fauna (point categories) shown. Note that cod were not marked, despite being very abundant.

stones. All other individuals of these groups of animals were located on crusts in mosaic A, with the exception of white, rounded sponges, which were also located on soft sediment (122 were on crusts).

Sulfide was undetectable in all push cores at the sediment-water interface and usually in the first few centimeters (Fig. 6, Table S1). However, in three cores, sulfide was detectable and present in high μM concentrations deeper in the sediment. In core P18-028, sulfide reached 176 μM at the depth of 16 cm, and continued to increase with increasing depth, reaching nearly ten times this concentration at the base of the core (33 cm depth). In core P18-031, sulfide was detectable (43 μM) at shallower depths in the sediment (10 cm depth), with a much lower peak concentration (327 μM), at 17 cm depth. Below 17 cm, sulfide concentration decreased, though remained present in appreciable amounts (168 μM). Sulfide was not detectable in core P18-033 until a depth of 29 cm and at this depth, concentration was still quite low (1.6 μM). Deeper in the core, concentration increased, reaching a maximum of 273.5 μM at the bottom of the core (45 cm depth). Sulfide was not detectable over the entire length of core P18-032. Decreases in sulfate concentrations roughly paralleled the trends of sulfide increases, though with a much smaller range (Fig. 6, Table S1).

**Table 1  Community details of the seep and non-seep sites.** Total numbers of individuals, densities, areas and numbers of polygons of the different faunal groups and features marked in the two mosaics. Features and animals marked as areas or polygon feature classes are represented with a + sign. Animals and features marked individually, as point feature classes, follow below. Density of individuals and polygons are calculated as per m², based on the total area of the mosaic, which itself is the first entry in the table. In addition to the fauna listed here, cod (*Gadus morhua*) were seen in large numbers in the two mosaicked areas.

| fauna/feature | number of individuals/polygons | | density of individuals/polygons | | total area of polygons | |
|---|---|---|---|---|---|---|
| | mosaic A | mosaic B | mosaic A | mosaic B | mosaic A | mosaic B |
| total area | 832.41 | 267.25 | | | | |
| microbial mats+ | 3 | 0 | 0.004 | 0 | 1.01 | 0 |
| seep crusts+ | 182 | 0 | 0.22 | 0 | 107.11 | 0.02 |
| cobbles and boulders+ | 73 | 2 | 0.09 | 0.01 | 2.86 | 0 |
| all rock features+ | 255 | 2 | 0.31 | 0.01 | 109.97 | 0.02 |
| fossilized worms+ | 21 | 0 | 0.03 | 0 | 3.65 | 0 |
| unknown sediment organisms+ | 153 | 39 | 0.18 | 0.15 | 764.30 | 112.35 |
| anemones | 21 | 0 | 0.03 | 0 | n/a | n/a |
| corals | 52 | 0 | 0.06 | 0 | n/a | n/a |
| seastars | 54 | 0 | 0.06 | 0 | n/a | n/a |
| *Sebastes* sp. (rockfish) | 6 | 1 | 0.01 | 0 | n/a | n/a |
| encrusting sponges | 361 | 19 | 0.43 | 0.07 | n/a | n/a |
| white round sponges | 269 | 0 | 0.32 | 0 | n/a | n/a |
| yellow sponge | 111 | 0 | 0.13 | 0 | n/a | n/a |
| holes in sediment | 1715 | 540 | 2.06 | 2.02 | n/a | n/a |
| unknown | 21 | 0 | 0.03 | 0 | n/a | n/a |
| total living fauna (point classes) | 895 | 20 | 1.08 | 0.07 | n/a | n/a |

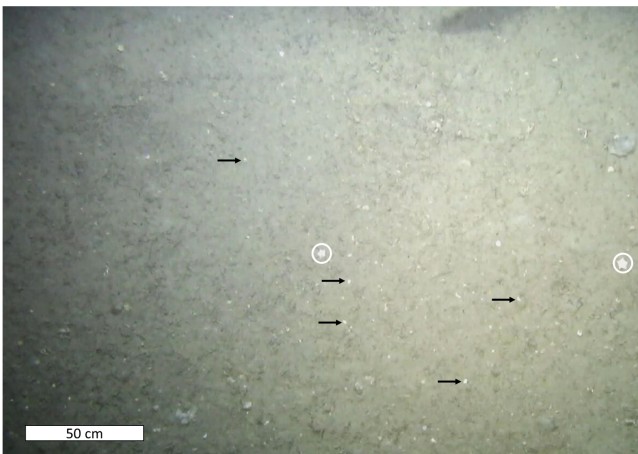

**Figure 4  White sediment organisms.** Sediment containing abundant small, white animals that could not be identified with the images. The animals, or body parts appear as small, white pinpoints within the sediment and a few are highlighted with black arrows. Note the sea stars in this image (circled in white).

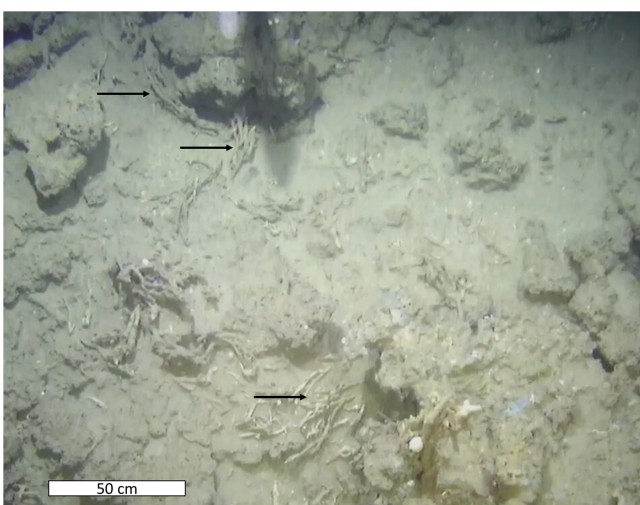

**Figure 5 Possible fossilized worms among crusts.** Features seen among some of the crusts that could be fossilized worms (though the phylum is not possible to identify). Black arrows point to a few aggregations.

## DISCUSSION

### Seep morphology in relation to biota

The oasis effect of cold seeps, particularly at depths where photosynthesis is not possible is well documented (*Sibuet & Olu-Le Roy, 1998*; *Sibuet & Olu-Le Roy, 2002*; *Fisher et al., 2007*; *Kiel, 2010*; *Levin et al., 2016*). However, the oasis effect is often considered to refer solely to abundance and biomass. Diversity and species richness, on the other hand, are often low at seep sites, which is usually attributed to the toxicity of reduced compounds such as sulfide to most life forms. At northern latitudes though, qualitative evidence exists suggesting that seeps are more speciose in terms of megafauna than the background benthos, in addition to hosting higher abundance (*Gebruk et al., 2003*; *Sen et al., 2018a*). We tested this concept quantitatively at a seep site earmarked for hydrocarbon drilling in the southwestern Barents Sea. It should be kept in mind that due to significant time constraints, only one mosaic was obtained for each of the two habitats in question (seep and non-seep background). Admittedly, a sample size of 1 is inadequate for drawing robust conclusions. However, our comparisons are not between samples in the true sense, rather, we are comparing large areas of the seafloor (832.4 $m^2$ and 267.3 $m^2$ for mosaic A and B respectively) which does allow for some insight.

Our results indicate both higher abundance and number of megafaunal taxa at the seep than the background site (Figs. 2–3, Table 1). In fact, the non-seep seafloor was so homogeneous and devoid of megafaunal species that the mosaicking process, which requires conspicuous common features between successive images for blending purposes, was difficult to carry out. With the exception of cod (*Gadus morhua*), who showed a tendency to follow the ROV, the only identifiable living fauna in mosaic B were some encrusting sponges (in considerably fewer numbers than mosaic A of the seep area), and one individual of *Sebastes* rockfish.

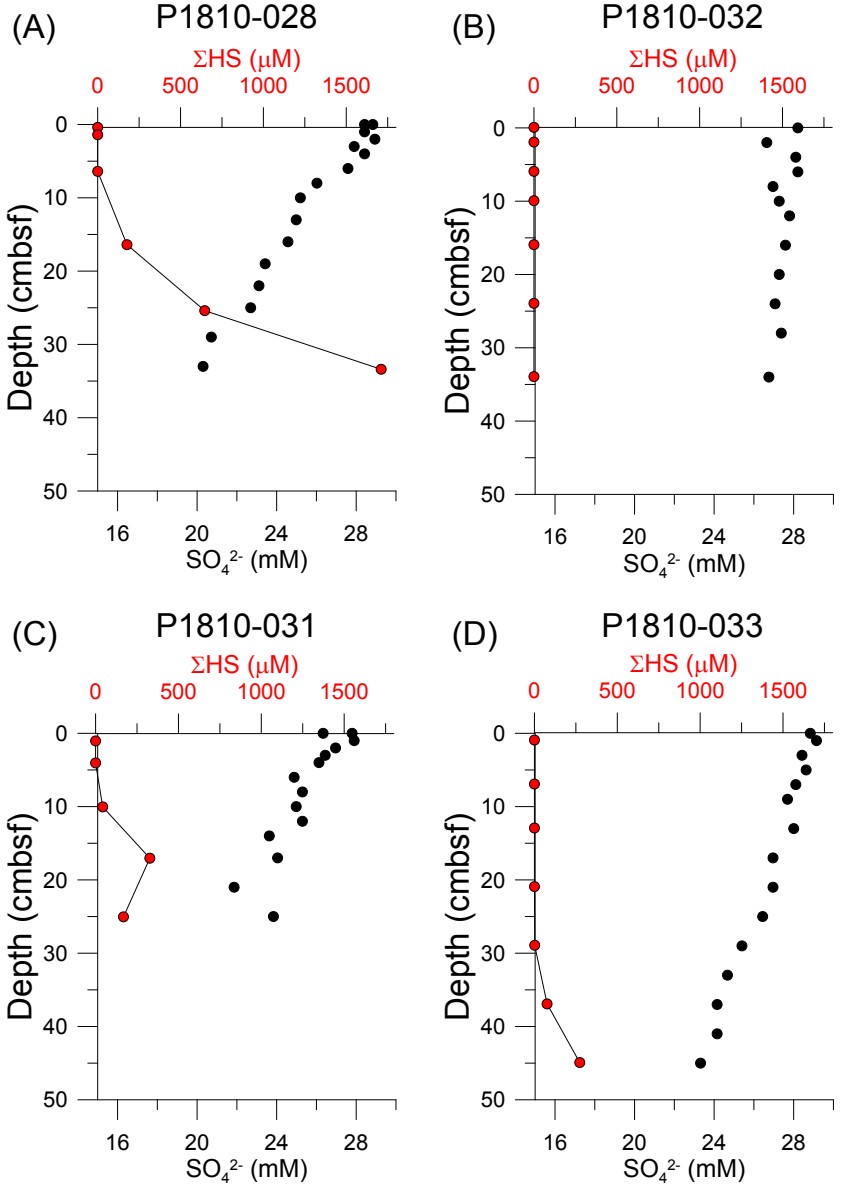

**Figure 6** **Sediment geochemical profiles.** Porewater concentrations of sulfide and sulfate measured from push cores taken from locations within mosaic A. (A) P1810-028, (B) P1810-032, (C) P1810-031, (D) P1810-033.

Therefore, at Svanefjell, within mosaicked areas, it appears that more megafaunal species or taxa aggregate around the seep site than the non-seep, background seafloor. Abundance is also clearly higher as well (Table 1) and though we have no measurements, biomass at the seep site likely also exceeds the biomass of the background community. In fact, the one coral in the center of mosaic A alone likely constitutes a higher biomass than the entire visible megafaunal community in mosaic B (Fig. 2).

Crusts of seep carbonates appear to be a major factor contributing to the higher megafaunal diversity and abundance at the Svanefjell seep site (compared to the non-seep location). These crusts are formed due to microbial activity that increases carbonate alkalinity in the sediment, specifically, through the anaerobic oxidation of methane (AOM) coupled to the reduction of sulfate (*Boetius et al., 2000*; *Joye et al., 2004*; *Arvidson, Morse & Joye, 2004*; *Knittel & Boetius, 2009*). They can persist after seepage ceases and reservoirs are exhausted, and provide settlement surfaces for a variety of hard substrate dwelling animals including reef building corals (*Cordes et al., 2005*; *Cordes et al., 2008*; *Becker et al., 2009*; *Lessard-Pilon et al., 2010*). Most of the animals in mosaic A were colonizing crust surfaces. Only seastars, fish and the masses of small, white, unknown fauna in the sediment comprised the non-hard substrate dwelling community members within mosaic A (some sponges were seen both on the crusts and in the sediment). Therefore crusts account for nearly all of the fauna in mosaic A. The absence of crusts from mosaic B subsequently likely accounts for the extremely sparse megafaunal community in the background site. Other than a hard settlement surface, another utilitarian feature of crusts is their irregular morphology and the presence of numerous cavities which can be used for protection or concealment. Therefore, even non-encrusting or non-hard substrate dwellers were seen around crusts in mosaic A, such as the red rockfish (*Sebastes* sp.). Only one such individual was seen in mosaic B, where the only hard, rock like features were cobble size stones that are likely erosional or glacial remnants. In fact, the potential for crusts of seep carbonates to reach large sizes, across all dimensions could be significant and indeed, the vast majority of animals in mosaic A were seen colonizing crusts, while very few were observed on stones and rocks. This is probably due to the small sizes of stones in the mosaic: they simply do not have the surface area to support many encrusting or hard substrate megafaunal species. In addition to the provision of relatively large hard surfaces, crusts of seep carbonates also create more heterogeneity than stones which on average are smaller, or at least more discrete units. This leads to animals such as rockfish aggregating around them, which does not take place to the same extent around smaller stones. Therefore, crusts of seep carbonates not only provide hard settlement surfaces, but additionally provide them in large enough quantities. This feature is likely critical to them supporting and enhancing local abundances of animals including hard substrate dwelling species.

The presence of hard substrates such as rock features and boulders on the seafloor has been demonstrated to increase local diversity due to their ability to provide settlement surfaces and refuge, and because elevated positions off the seafloor can entrain currents leading to enhanced food supply (*Hargrave, Kostylev & Hawkins, 2004*; *Duineveld et al., 2007*; *Felley, Vecchione & Wilson, 2008*; *Becker et al., 2009*; *Post et al., 2017*). In high latitude regions, glacial ice rafted debris of varying sizes (dropstones), constitute an important diversity enhancing hard substrate on the seafloor (*MacDonald et al., 2010*; *Schulz et al., 2010*; *Meyer et al., 2016*; *Post et al., 2017*; *Ziegler et al., 2017*). Species richness of communities associated with dropstone 'island' habitats has been shown to correlate with size of the dropstones (*Meyer et al., 2016*; *Ziegler et al., 2017*) and the presence of these rock features can serve as a means of dispersal for hard substrate animals (*Post et al., 2017*). Crusts of seep carbonates likely function similar to dropstones and indeed,

our results suggest comparable trends, such as size and extent correlating with diversity or richness. A significant difference between seep carbonates and dropstones is that the former are more likely to be present in relatively high concentrations where they are present, whereas the latter tend to be more dispersed and more randomly distributed in relation to each other. Therefore dropstones themselves function as islands, while seeps form larger units, with multiple instances of hard surfaces, interspersed among soft sediment. Subsequently, a single seep site can contain multiple 'islands' and would therefore be more analogous to an archipelago, and it remains to be seen how or even whether the trends of island biogeography that have been observed among dropstone habitats (*Meyer et al., 2016*; *Ziegler et al., 2017*) applies to crusts of seep carbonates. The random occurrences of features such as dropstones means that our results, of a seep containing more taxa in larger numbers than the surrounding seafloor is to a certain extent, a product of the local characteristics of the study area. For example, had a major rock outcrop, or numerous dropstones been present in the mosaicked background area, we might not have recorded such a big difference between the seep and non-seep site. The uneven distribution of animals and the possibility of species to be highly concentrated in certain locations needs to be considered. Nonetheless, certain habitats, such as coral reefs, eelgrass beds, sponge gardens and kelp forests on average tend to be more diverse and more abundant in fauna than the background seafloor (*Tendal, 1992*; *Buhl-Mortensen & Mortensen, 2004a*; *Buhl-Mortensen & Mortensen, 2004b*; *Buhl-Mortensen et al., 2010*; *Ottersen et al., 2011*; *Gonzalez-Mirelis & Buhl-Mortensen, 2015*). Our results, in addition to other studies that have made comparisons between seep sites and non-seep areas in high latitude regions (*Gebruk et al., 2003*; *Åström et al., 2018*; *Sen et al., 2018b*) suggest that seeps might similarly function as local diversity hotspots, and that crusts of seep carbonates factor strongly in their ability to enhance local diversity.

## Seep geochemistry in relation to biota

The main factor driving the aggregating effect at the Svanefjell seep site appears to be the presence of crusts of seep carbonates, as opposed to food availability and local production. In fact, sulfide was undetectable at the sediment-water interface in all four sampled push cores and only reached detectable concentrations deeper in the sediment (Fig. 6). This suggests that even though seepage is taking place, it is in low amounts, and the site overall might have experienced a decline in seepage overall since neither visual nor echosounder based observations of gas flares were recorded during this study, despite both having been recorded before. Subsequently, siboglinid frenulate worms were absent. These worms are the dominant community members of high latitude seeps (*Smirnov, 2000*; *Gebruk et al., 2003*; *Rybakova Goroslavskaya et al., 2013*; *Paull et al., 2015*; *Åström et al., 2016a*; *Åström et al., 2018*; *Sen et al., 2018b*; *Sen et al., 2018a*; *Sen et al., 2019*) and through their symbiotic association with chemoautotrophic bacteria, form the base of the food chain within their habitats. Sulfide has been demonstrated to be the energy source for high latitude seep frenulates (*Lösekann et al., 2008*; *Sen et al., 2018b*) and though these worms reach 50–60 cm in length (*Gebruk et al., 2003*; *Sen et al., 2018a*; *Sen et al., 2018b*) and likely acquire sulfide across their bodies in the sediment similar to vestimentiferan siboglinids (*Freytag*

*et al., 2001*), quantities might still be insufficient to sustain them at the Svanefjell seep site. The absence of sulfide in core P1-032 (at least to the point of our detection limit of 0.2 μM) suggests that whatever sulfide is present at the site is extremely patchy, which likely makes colonization and growth of frenulates difficult. Sulfide flux, in addition to concentration has also been suggested to determine the presence or absence of frenulates; even when sulfide is abundant, frenulates can be absent if fluxes are low (*Sen et al., 2018a*). Though we did not measure sulfide flux at the Svanefjell site, the sulfide and sulfate data point towards low fluxes of reduced compounds. Methane has also been suggested to be an energy source for frenulates from a seep site in the Laptev Sea (*Savvichev et al., 2018*), however, since sulfide at seeps is produced from the anaerobic oxidation of methane, it is likely that methane availability parallels that of sulfide, and is present deeper in the sediment, and patchily, at the Svanefjell seep.

Interestingly, frenulates occur in non-seep environments and have been hypothesized to utilize insoluble sulfides when sediment dissolved sulfide concentrations are very low (*Dando et al., 2008*). However, different species have also been linked to different sediment geochemical conditions (*Sen et al., 2018b*); therefore, it is possible that despite conditions at the Svanefjell seep being suitable for some frenulate species, they are nonetheless inadequate for the *Oligobrachia* frenulates that populate high latitude seep sites. In short, the low levels of seeping of hydrocarbons and subsequent low concentrations of sediment dissolved sulfide in the first 10–30 cm probably accounts for the absence of chemosymbiotrophic frenulate worms.

A major contributor towards seep primary production is therefore missing at the study site. Additionally, microbial mats were barely present (in line with sulfide being undetectable at the sediment-water interface in all our sampled cores), and together this suggests that food production is relatively low overall. This could account for the generally low abundance and number of taxa at the Svanefjell seep (even if they are higher than the background area). At comparative depths, highly active seep sites with sulfide reaching mM concentrations in sediment porewater have been seen to host highly diverse (60+ taxa) and extremely abundant communities of animals (*Sen et al., 2018a*). The draw of seep sites to benthic fauna is basically twofold: seeps provide more (in comparison to the background seafloor) food and hard substrates. In the case of the Svanefjell site, the former factor, i.e., increased food availability and productivity, is likely diminished because of low levels of seepage of hydrocarbons from sub-seabed reservoirs. As a result, the Svanefjell seep is scarce with respect to numbers of taxa and individuals of megafauna in comparison to other seep sites, and whatever aggregating effect it might have with respect to megafauna is likely concentrated around the presence of crusts.

Nonetheless, primary production at the study site might likely contribute somewhat to its relative oasis effect. Nearly the entire mosaic A was covered in small, white organisms that could not be identified at the scale of the images, but are possibly small cnidarians or sponges, or polychaetes (Fig. 4). These animals were present, but were considerably less abundant in mosaic B (less than half of mosaic B vs more than 90% of mosaic A). Without sediment samples, it is impossible to determine what these animals are, however, their increased presence in the seep site compared to the background site suggests that food

production at the seep could draw certain taxa to the site, since these animals were present on soft sediment. Seepage and concentrations that are undetectable to our equipment could nonetheless fuel autotrophic processes at the seep, both for free living microbes and possibly, within infaunal animals such as thyasirid bivalves that have been recorded at high latitude seeps (though symbioses in these species have not yet been determined) (*Åström et al., 2016b*; *Åström et al., 2018*). Furthermore, reduced chemical compounds serving as energy sources are present deeper in the sediment, as demonstrated by the increases in sulfide seen in three of our sampled cores. Therefore, though it was beyond the scope of this study to demonstrate it, it is very likely that the microbial community at the seep and the background site are significantly different, and that the rates of processes such as AOM are higher at the seep than outside. These rates might decrease towards the sediment surface, in correspondence with the availability of reduced compounds, which is probably why visible microbial mats are barely present at the seafloor of the seep site. Subsequently, energy transfer up the food chain is likely limited, resulting in smaller megafaunal communities than more active seep sites. However, considerable energy and carbon transfer is probably still taking place: meiofaunal and macrofaunal sediment communities at the seep are likely to host higher biomass than these communities outside the pockmark, and the visual manifestation of this is the abundance of small white animals in the sediment all over mosaic A.

## Implications of seepage for biota and marine management

Our results reveal that even low level seepage can enhance both the number of taxa and their abundance on the seafloor, at least in the northern latitude setting of our study. Crusts of seep carbonates account for most of this effect, which might not be related whatsoever to current seepage patterns since crusts take hundreds to thousands of years to form, and U-Th dating has demonstrated that the main seepage and crust formation episode in the SW Barents Sea occurred between 16 and 9 ka ago, shortly after the deglaciation of the area (*Crémière et al., 2016a*). However, local chemosynthesis based food production might contribute to the effect as well and the degree to which seeps function as local oases and hotspots of diversity and biomass probably correlates with how active a particular seep is. For example, seeps with higher hydrocarbon flux and correspondingly higher concentrations of reduced gases in sediment porewater sustain significantly more diverse and abundant communities than the study site (e.g., *Rybakova Goroslavskaya et al., 2013*; *Sen et al., 2018a*) and therefore, enhance local seafloor diversity and abundance even more. This means that the extent to which seeps enhance local diversity and abundance changes during its life cycle, as does the manner of its influence. Young, active seeps with plenty of fluid discharge probably sustain abundant chemosynthesis based organisms, including siboglinids and likely contain large areas covered by microbial mats. Grazers and higher order organisms in turn aggregate at these sites, but encrusting animals and hard substrate dwellers likely only appear later, as crusts form. Active seeps where emissions have been occurring for sufficient time to allow for large crust deposits to form likely represent the pinnacle at which seeps attract and host benthic fauna. As seepage wanes, chemosynthesis based organisms decrease, and the aggregating effect of seep sites is mainly limited to the

ability for crusts to provide substrates, shelter and heterogeneity. Even at this stage though, seep sites appear to be more diverse and appear to host higher abundances of benthic fauna than the surrounding seafloor. Confirming whether these trends hold true requires much more thorough and comprehensive surveys and characterization of seep sites and it should be kept in mind that these trends might be affected by water depth, since seeps overall tend to display distinct patterns with respect to diversity and water depth (*Sibuet & Olu-Le Roy, 1998*; *Sahling et al., 2003*; *Dando, 2010*; *Sen et al., 2018b*; *Sen et al., 2018a*).

It is important to note that seeps in northern latitude settings have to date, been seen to host general, benthic fauna, and seep endemic or specialist species are notably scarce or absent. Even the chemosymbiotic siboglinids of northern seeps are under debate as being a seep specific species, and might correspond to a fjord species (*Sen et al., 2018b*). In fact, the only faunal group observed at high latitude seeps that are potentially seep specific are rissoid snails, which have been observed at seep sites in the Norwegian Sea (*Decker & Olu, 2012*; *Decker et al., 2012*). *Sen et al. (2019)* hypothesized that conditions in polar regions might be responsible for this trend; specifically that seeps, with their light-independent primary production, shield against highly reduced food input from the surface during dark winter months. They provide a haven for benthic organisms when other areas of the seafloor are experiencing particularly enhanced deprivations of organic matter deposition from surface waters. Subsequently, Arctic and high latitude benthic fauna aggregate at seeps and over lengthy time spans, and as a result, seep communities in the northern parts of the world consist primarily of regular, 'background' species as opposed to specialist fauna. It has even been suggested that this scenario has led to certain background species possessing elevated tolerances for the toxic, sulfidic conditions of seeps primarily seen among seep specialist fauna in lower latitude regions (*Sen et al., 2019*). Therefore, seeps could play a significant role in the larger marine ecosystem in high latitude regions and their aggregating effect could be an important factor in benthic ecosystem dynamics.

The fact that both current (chemoautotrophic primary production) and past processes (crust formation) at seeps can result in them functioning as a beacon for benthic fauna is an important consideration for ecosystem management, since the sub-seafloor reserves that fuel seepage are highly valuable commercial resources, as are some of the aggregating animals, such as cod. Further examination is necessary to assess the scale and nature of the role seep sites play in northern benthic ecosystems and this evaluation is likely to now be quite time sensitive, given the rapidity with which northern regions are experiencing changes. Whether or not the aggregating effect of seeps can be transferred to them serving as refuge sites for northern species as southern species shift their ranges northward is also pertinent. Our results suggest that seeps are important components of marine ecosystems and in northern regions, the links might be particularly strong, or at least relevant across all kinds of species and taxa rather than through a relatively small number of somewhat specialized fauna. Importantly, senescent seeps or sites with weakening levels of seepage can still have heightened benthic diversity and abundance and this effect can continue long after the complete cessation of seepage and 'death' of a seep site since crusts alone can account for the aggregating effect of seeps. Furthermore, the spatial extent of the influence of seeps might not be limited to small areas as discussed here. It has been suggested that

the presence of large coldwater coral reefs in Norwegian waters is linked to nearby sites of leaking hydrocarbons (*Hovland & Thomsen, 1997*; *Mortensen et al., 2001*; *Jensen et al., 2008*; *Jensen et al., 2012*; *Jensen et al., 2015*; *Hovland, Jensen & Indreiten, 2012*; *Hovland, 2012*).

In fact, the significance of seeps to larger marine ecosystems and even to global processes is increasingly being recognized. Carbon sequestration and cycling, regulation of methane fluxes, oxygen consumption, nutrient cycling, production export, enhancing heterogeneity on continental margins etc., are some of the commonly discussed topics within the context of seep ecosystem services and impacts (*Cordes et al., 2010*; *Le Bris et al., 2017*; *Levin et al., 2016*). However, major gaps still remain with respect to the effects of seeps and the quantitative importance of their chemosynthesis based production at regional scales (*Le Bris et al., 2017*). In high latitude locations this deficiency is particularly striking. For a considerable length of time, studies on high latitude seeps were largely restricted to the Haakon Mosby mud volcano (e.g., *Gebruk et al., 2003*; *De Beer et al., 2006*; *Niemann et al., 2006*; *Lösekann et al., 2008*; *Rybakova Goroslavskaya et al., 2013*; *Smirnov, 2014*). Only recently have other northern and Arctic seeps been subjected to detailed study. Much of this research, including this study, has demonstrated or postulated the impacts of seeps to northern marine ecosystems (*Åström et al., 2016b*; *Åström et al., 2018*; *Sen et al., 2018a*; *Sen et al., 2019*). Since northern regions and the Arctic contain vast marine methane reserves, the connections between seeps and the larger marine ecosystem might be particularly pronounced in northern regions, and the impact of seeps could be particularly widespread. One expected impact of the extensive marine methane reserves of the Arctic was a disproportionate effect on changing climatic trends; an extension of this line of thought would suggest that northern methane based systems such as seeps would additionally play a significant role in global geochemical cycles and processes. This means that quantifying links and impacts of high latitude seeps and the encompassing ecosystem hold significance at both the regional and global scale. Therefore, governments or policy makers in northern latitudes need to consider the role of seeps quite closely and balance scientific knowledge with economic needs to maintain and use these habitats that hold considerable value from both a commercial and an ecological perspective.

## CONCLUSIONS

Studies have suggested that the common generalization of seeps exhibiting high biomass/abundance, but low diversity in comparison to the surrounding seafloor, does not hold true in high latitude settings. We provide quantitative evidence of this: through image based seafloor mapping we demonstrate that the Svanefjell seep site in the southwestern Barents Sea hosts both higher abundance and number of megafaunal taxa than the adjacent non-seep seafloor. Crusts of seep carbonates account for most of the enhanced taxonomic richness and abundance of megafauna at the seep site, however, primary production might play a role as well, and probably is highly significant with respect to sediment fauna at both the macrofaunal and meiofaunal scale, although this could not specifically be tested in our study. Despite higher richness and abundances compared to the background benthos,

the Svanefjell site hosts considerably fewer taxa, and in lower abundances than other high latitude seeps at comparable water depth. Low levels of seepage likely account for this, and low seepage was confirmed through porewater sediment chemistry analyses. Our results, in combination with other studies suggest a significant role of cold seeps in the marine benthic ecosystem of northern latitude regions: they possibly function as local diversity and abundance hotspots. The extent and degree of this effect is likely tied to the life cycle of the seep system itself, and a function of fluxes and seepage activity. Notably, however, even waning stages enhance local diversity and abundance and this trend can continue after the 'death' of a seep, since seep crusts are one of the major factors contributing towards the draw of seep sites to benthic species. Therefore, modern day sites of active seepage as well as locations where seepage once existed in the past both likely are closely tied to the benthic ecosystem, and exploring this concept in more detail is necessary for a more holistic understanding of northern latitude marine systems.

## ACKNOWLEDGEMENTS

We are extremely grateful to the captain and crew of the *Bourbon Arctic* and to Dennis Örhill and the Bourbon DNT (ROV) crew, without whom this project would not have been possible. We are thankful to Carl Ballantine for assistance at sea and to Mats Andersson, who helped construct Fig. 1. We thank Dr. Magdalena Georgieva and Dr. Crispin TS Little for attempting to identify the 'fossilized worms' among crusts. Dr. Alix Post and Dr. Erik Cordes provided valuable comments and suggestions that helped refine this manuscript.

### Funding

This project was funded by Lundin Norway. Postdoc support for AS was provided through the Centre for Arctic Gas Hydrate, Environment and Climate (CAGE) and the Research Council of Norway through its Centres of Excellence scheme (grant number 223259). The funders had no role in study design, data collection and analysis, decision to publish, or preparation of the manuscript.

### Grant Disclosures

The following grant information was disclosed by the authors:
Lundin Norway.
Centre for Arctic Gas Hydrate.
Environment and Climate (CAGE).
Research Council of Norway through its Centres of Excellence scheme: 223259.

### Competing Interests

Rolando di Primio and Harald Brunstad are employees of Lundin Norway.

## Author Contributions

- Arunima Sen conceived and designed the experiments, performed the experiments, analyzed the data, contributed reagents/materials/analysis tools, prepared figures and/or tables, authored or reviewed drafts of the paper, approved the final draft.
- Cheshtaa Chitkara analyzed the data, approved the final draft.
- Wei-Li Hong analyzed the data, contributed reagents/materials/analysis tools, prepared figures and/or tables, approved the final draft.
- Aivo Lepland conceived and designed the experiments, analyzed the data, contributed reagents/materials/analysis tools, approved the final draft.
- Sabine Cochrane approved the final draft, and was chief scientist on the research cruise.
- Rolando di Primio and Harald Brunstad conceived and designed the experiments, approved the final draft.

## Field Study Permissions

The following information was supplied relating to field study approvals (i.e., approving body and any reference numbers):

This study was carried out with the approval of Lundin Norway (Production License 934).

## Data Availability

All data used in this manuscript are available in the article and the Supplemental Material.

## Supplemental Information

Supplemental information for this article can be found online at http://dx.doi.org/10.7717/peerj.7398#supplemental-information.

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
