# Peer review of "Image based quantitative comparisons indicate heightened megabenthos diversity and abundance at a site of weak hydrocarbon seepage in the southwestern Barents Sea"

_PeerJ, doi:10.7717/peerj.7398_

## Round 0.1 · original submission · Minor Revisions

Both reviewers are positive about the paper overall and complimentary regarding your approach to the subject. Reviewer 1 has provided numerous editorial suggestions that I agree are needed to improve the paper. Reviewer 2 suggests some further review of existing video data to better explain congregations of mobile fauna - if you have no other data that could address this please explain this in the paper. I ask that you attend to all corrections and carefully consider the suggestions of both reviewers in making minor revisions.

·

Basic reporting

This paper provides an interesting perspective on the interaction between benthic megafauna and the characteristics of a hydrocarbon seepage site. Importantly, the authors find a strong association between the occurrence of hard substrates, in the form of carbonate crusts formed due to the seepage at this site, and the abundance and diversity of the benthic fauna. As the authors note, this is an important finding that highlights the significance of current or former seepage for understanding the nature of the benthic ecosystem. This paper adds to our knowledge of habitat heterogeneity associated with hydrocarbon seeps, and the implications for managing this environment.

Language – This is mostly of a high quality throughout. I have made some suggested edits in the General Comments below.

Introduction and background – these are well written and set the context well. Relevant references are provided. One area that could be expanded from the literature is a more general discussion of habitat heterogeneity due to variations in substrates, and its impact on benthic diversity and abundance. I have noted this below.

Structure – The overall structure is appropriate, but the addition of sub-headings to the discussion would help to provide a more coherent structure to this section. I suggest that the Discussion be rearranged under the following headings: i) seep morphology in relation to biota; ii) seep geochemistry in relation to biota; iii) implications of seepage for biota and marine management. There are also some results in the Discussion section that should be moved, and vice versa. I have noted these under General Comments below.

Figures – Figures are all relevant, but I do have some suggested edits. The authors note observations of gas flares from the multibeam data and bubbling gas in the ROV imagery. It would be useful to show this data and identify the location of these observations on Figure 1.
Figure 1 (top) – add a label for the Barents Sea
Figure 1 (bottom) – Add an outline to show the exact locations and extent for the Area A and B mosaics, and the observed gas flares and gas bubbles.
Figure 2 – Some of the symbols are very difficult to see on this image. Small polygons can not be easily distinguished from the crosses, especially since some of the colours for the polygons and crosses are very similar. The red and pink for the anemones and Sebastes sp. can also not be distinguished. It’s confusing having a mixture of crosses and polygons for the substrate and biological data e.g. sediment holes are crosses, but all the other crosses are biological; unknown organisms and microbial mats are polygons, but the rest of the polygons are for substrate characteristics. This needs to be presented more clearly, maybe with a separation between substrate and biological data into different figures.
Figure 3 – I cannot see the stone/rock polygons on this figure.
Figure 4 – Place arrows to identify seastars and different arrows (black vs white perhaps) to point to a few examples of the unidentified white biota.
Figure 5 – Although fairly evident, place a few arrows on this to highlight fossilised worms.
Table 1 – Add total density of fauna to this. For the individual counts, place the density first, with the number of observations in brackets for consistency to the order in which the polygon data is presented.
Raw data – This is supplied appropriately.

Experimental design

This is original primary research that is appropriate to the scope of PeerJ. The methods are clearly explained and appropriate. The research makes a useful contribution to our understanding of seafloor ecosystems, and their association with hydrocarbon seepage features. Further analysis of the nature of the microbial communities between the seepage and non-seepage sites would strengthen this study and limit the speculative nature of some of the discussion (e.g. from line 358). I wonder whether this would be possible based on the sediment cores collected? It is a shame that no cores were collected from Area B. However, the study is still useful in terms of understanding the habitat heterogeneity associated with seepage sites, and the influence that this has on the benthic ecosystem.

Validity of the findings

The analysis that is presented is appropriate for the data. With only 2 sites available the authors acknowledge that only general conclusions can be drawn, and their conclusions are well-constrained by the data. While the journal does support speculation, I found that there was too much speculative discussion regarding the significance of the unidentified organisms (Line 363-382). I also think that the authors need to reconcile the evidence for active seepage, such as the flares seen on the multibeam and the bubbles on the ROV images, with the absence of detectable sulphides in the upper layers of the sediment.

Additional comments

Line 111, 112, 118 – Do you have references to the seismic data, multibeam gas flares and bubbling gas that can be referred to here, or can you present any of this data?
Line 124 – what was the speed of the ROV across the seafloor?
Line 125 – replace text to “lawn mower fashion, to provide complete coverage, ensuring overlap between successive lines.”
Line 167 – replace text to “seen in mosaic A (Figs. 2-3, Table 1), while mosaic B…”
Line 169 – delete “could be animals that could not be identified at this level of resolution, but”
Line 171 – rewrite “The patches of small…”
Line 172 – Remove “However”, rewrite “Holes in the sediment were observed in similar quantities in both mosaics”
Line 177 – The addition of this animal at both the seep and background site increases the taxonomic richness by the same amount at both sites. The wording of this line implies that it only increases the richness at the non-seep site.
Line 182-184 – This belongs in the Discussion, not results
Line 187 – replace “what could be” with “possible”
Line 194 – replace “end” with “base”
Line 194 – “core P18-033” – do you mean core P18-031 here?
Line 194-195 – remove “the point at which”
Line 195-197 – rewrite as “detectable (43 um) at shallower depths in the sediments (10 cm depth), with a much lower peak concentration (327 um) at 17 cm depth. Below 17 cm sulphide concentration decreased, though remained…”
Line 198 – replace “till” with “until”
Line 199 – Delete “but”, rewrite as “(1.6 uM). Deeper…”
Line 202 – replace “they were not nearly as dramatic” with “though with a much smaller range”
Line 227-235 – this is repetitive of the results already presented
Line 237-242 – this belongs in the Results.
Line 239 – remove “ourselves”
Line 240 – rewrite as “seep site also likely exceeds the…”
Line 245 – replace “location in which the study was conducted” with “local characteristics of the study area”
Line 242-244 – It would be good to include a broader discussion and context of how seafloor heterogeneity in substrates influences the distribution, diversity and abundance of the seafloor fauna. Relevant examples for high latitudes would be literature relating to the distribution of dropstones e.g. Schulz et al., 2010; Post et al., 2017; Ziegler et al., 2017.
Lines 250-252 – The relevance of these examples to this study is not clear.
Line 255 – The comment that “seeps might function in a similar fashion as well” needs to be explained.
Paragraph 237-255 would be better placed in the context of the paragraph that follows. I think restructuring the Discussion as I have suggested above may assist in making a more coherent flow.
Line 276-287 – This belongs in Results
Line 294 – Full stop after “quantities”. Remove “and”. Start new sentence.
Line 297 – remove “Therefore, in this case”. Replace “of the” with “at this”
Line 331 – Remove “This would mean that”. Add “therefore” after “production is”
Line 369 – Remove “For example,”. Replace “ought” with “should”
Line 363-382 – This speculative discussion could be summarised by adding a sentence to Line 230 to mention the significance if these organisms are thyasirid bivalves. But given there is no certainty that these are bivalves, or seep-specific organisms, this discussion seems unnecessarily detailed.
Line 390 – The statement “local chemosynthesis based food production also contributes to the effect as well” is too strong given that this study does not provide conclusive evidence of a dependence of any of the organisms on a chemosynthetic food source.
Line 394 – add a reference after “diverse and abundant communities” to back up this statement. Also, this statement seems contrary to what is stated in other sections that sites of active seepage have relatively low diversity (e.g. line 81, 87-88).
Line 404-405 – it would be good to see further context here of whether the increase in abundance and diversity associated with the carbonate crusts is similar to that associated with other types of random, relatively small occurrences of hard substrates at high latitudes (e.g. dropstones).
Line 412 – Replace “till” with “to”
Line 416 – Replace “that” with “which”
Line 412-427 – In your argument that seeps may provide an alternative food source in regions with highly seasonal primary production it is not clear how the non-seep fauna observed would be able to adapt to the toxic nature of seeps in utilising these areas as a food source. This needs to be addressed.
Line 463 – Replace “compared to” with “than”

References cited:
Post, A. L., et al. (2017). "Environmental drivers of benthic communities and habitat heterogeneity on an East Antarctic shelf." Antarctic Science 29: 17-32.
Schulz, M., et al. (2010). "Colonisation of hard substrata along a channel system in the deep Greenland Sea." Polar Biology 33(10): 1359-1369.
Ziegler, A. F., et al. (2017). "Glacial dropstones: islands enhancing seafloor species richness of benthic megafauna in West Antarctic Peninsula fjords." Marine Ecology Progress Series 583: 1-14.

·

Basic reporting

This manuscript describes the relationship between the relatively shallow seep ecosystems of the Barents Sea and the surrounding deep sea. It joins a growing list of evidence for the significance of chemosynthetic productivity to the world's oceans. The fundamental conclusions of the manuscript are sound. It is a very well written manuscript, and it is refreshing to see something so polished at the initial manuscript submission stage.

Besides my minor comments listed below, my only major suggestion would be to expand the introduction and the end of the discussion to include more of the known links between chemosynthesis and the deep sea. There is a growing body of literature on the ecosystem services of chemosynthetic ecosystems and the significance of seeps to the surrounding deep ocean. A broader discussion of these links would be approrpiate here and would make this paper more broadly applicable and widely cited.

Experimental design

The experimental design is sound and the conclusions reached are well supported by the data. The only area that I would comment on here would be to go back to the ROV video and any other samples that are available to augment the data presented here. Larger scale photo transects could help resolve the difference between seep vs non-seep locations and the abundance of mobile fauna. Is there anything else in the background that causes these fauna to aggregate?

Video would also help in an attempt to get a better identification for some of the organisms and features observed in the photos. It is hard to believe that the only samples taken were the few push cores presented here. Even if they were not sampled, close-up video of some of the "white objects" should help to identify them. Thyasirids are discussed - is it possible that these are the white objects? Any other educated guesses? Likewise with the "fossil" tubes. It is possible that these are serpulids (either fossils or possibly even living worms), which can often be found near seeps.

Validity of the findings

No additional comments here.

Additional comments

In addition to the general comments above, a few other minor comments:

L309: Freytag et al. is not an appropriate citation here. Frenulates and vestimentiferans are very different organisms.

L446: While there may be some link between seeps and corals, these older Hovland papers have some fundamental issues. I would look to more recent studies.

---

## Round 0.2 · accepted · Accept

I have read the authors responses to the reviewers comments and am satisfied that the paper has been revised accordingly. No further revisions here needed.